# Clinical Results of Flow Diverter Treatments for Cerebral Aneurysms under Local Anesthesia

**DOI:** 10.3390/brainsci12081076

**Published:** 2022-08-13

**Authors:** Saujanya Rajbhandari, Hidetoshi Matsukawa, Kazutaka Uchida, Manabu Shirakawa, Shinichi Yoshimura

**Affiliations:** 1Department of Neurosurgery, Hyogo College of Medicine, Nishinomiya 663-8501, Hyogo, Japan; 2Department of Neurosurgery, Takarazuka City Hospital, Takarazuka 665-0827, Hyogo, Japan

**Keywords:** anesthesia, cerebral aneurysm, flow diverter, outcome

## Abstract

Flow diverters (FD) have become the mainstay for the endovascular treatment of unruptured intracranial aneurysms (UIA). Most FD procedures are performed under general anesthesia, and the influence of local anesthesia (LA) on outcomes remains unknown. This study evaluated the results of FD placement under LA. Data of patients treated for FD under LA between August 2016 and January 2022 were analyzed retrospectively. A good outcome was defined as a modified Rankin scale score of 0–2. Major stroke, steno-occlusive events of FD, mortality, and satisfactory aneurysm occlusion were also evaluated. In total, 169 patients undertook 182 treatments (139 [82%) female, mean age 61 ± 11 years). The median maximum aneurysm size was 9.5 mm (interquartile range 6.1-14 mm). A flow re-directed endoluminal device and pipeline embolization device were used in 103 (57%) and 78 (43%) treatments. One patient (0.59%) experienced major stroke, and steno-occlusive events were observed in four patients (2.4%). A good outcome at 90 days was obtained in 164 patients (98%), and one patient died (0.59% mortality). During the median follow-up period of 345 days (interquartile range 176–366 days), satisfactory aneurysm occlusion was observed in 126 of 160 aneurysms (79%). Our results suggest that FD placement under LA is a safe and effective treatment for UIA.

## 1. Introduction

Endoluminal flow diverters (FDs) such as the pipeline embolization device (PED [CovidieneEv3, Plymouth, MN, USA]) or Flow Re-Direction Endoluminal Device (FRED [MicroVention, Tustin, CA, USA]) have gained widespread acceptance for the treatment of intracranial aneurysms [1,2,3]. Most endovascular procedures, including FD placement, are performed under general anesthesia (GA) for patient comfort, control of hemodynamics and respiration, and better image quality due to decreased motion artifacts in immobile patients [4]. However, GA can carry a potential risk of respiratory and hemodynamic instability during induction, interfere with neurologic examination during the procedure, and have a longer total procedure time [5]. Patients undergoing FD deployment are placed on antiplatelet therapy to avoid thromboembolic complications. Few reports have also shown the risk of esophageal submucosal hemorrhage during the extubation of a tracheal tube as the patient is placed on antiplatelet therapy [6].

FD deployment can potentially cause vessel perforation, arterial thrombosis, dissection or stenosis, vasospasm, and perforator occlusion [7,8]. These complications can be preceded by neurological changes and, in GA patients, may go undetected until the procedure is completed, and the opportunity to intervene would have already passed.

Endovascular procedures are commonly performed in awake patients at our institution after the administration of local anesthetic agents and mild sedatives when required. To date, the influence of local anesthesia (LA) on clinical outcomes in the aneurysmal treatment of FD remains unknown. This study aimed to evaluate the clinical outcomes of FD placement under LA in patients with unruptured intracranial aneurysms (UIAs).

## 2. Materials and Methods

The study was reported based on criteria from the Strengthening the Reporting of Observational Study in Epidemiology statement [9]. The study protocol was approved by the Institutional Ethics Committee (No. 3229). Consent was obtained from all patients. Between August 2016 and January 2022, 170 patients with 173 UIAs underwent FD placement under LA at our institution. Overall, 17 patients with 18 aneurysms underwent the procedure under general anesthesia due to patient preference to undertake the procedure under GA because of restlessness, a difficult-to-access route (especially cervical portion), and initial FD introduction. Because one patient with one aneurysm underwent parent artery occlusion, this patient was excluded (Figure 1). Follow-up was conducted with magnetic resonance imaging/angiography (MRI/A) at 3 months, digital subtraction angiography (DSA) at 6- or 12 months, followed by annual MRA. Additional DSA was undertaken in case of suspected aneurysm filling. However, the routine follow-up was affected by the COVID-19 pandemic.

The clinical data were obtained through a review of the patients’ medical records and included demographic information, such as age, sex, body mass index, smoking history, daily alcohol consumption, and preoperative modified Rankin scale (mRS) [10]. The following data were also collected: duration of stay, previous treatment of aneurysm, medical history (hypertension, dyslipidemia, and diabetes mellitus), medications (statins and antiplatelet agents), platelet aggregability, P2Y12 reaction units (PRU), aspirin reaction units (ARU), aneurysm morphology (location, size, shape, and whether the aneurysm was thrombosed), type of FD, adjunctive coiling, and balloon angioplasty. Platelet aggregability was measured and classified as being from 1 to 9 using a PRP313M platelet aggregation analyzer (TAIYO Instruments, Osaka, Japan) 2 days prior to treatment, and ARU and PRU were measured using a Verify system (Accriva Diagnostics, San Diego, CA, USA) on the day of the procedure.

If platelet aggregability could not be suppressed sufficiently (ADP value 7–9), 20 mg of prasugrel or 200 mg of cilostazole was administered.

### 2.1. Outcomes

Major stroke (≥4 points on the National Institutes of Health Stroke Scale), steno-occlusive events of FD, mortality, and satisfactory aneurysm occlusion were evaluated. Satisfactory aneurysm occlusion was defined using the O’Kelly-Marotta grading scale C and D based on digital subtraction angiography, and no apparent aneurysm filling on magnetic resonance angiography [11,12]. A good clinical outcome was defined as an mRS score of 0–2. The mRS was evaluated at 90 days by telephone interviews with the patient or family members or during a physical examination in those who were able to visit our hospital.

### 2.2. Statistics

Statistical analyses were performed using JMP Pro (version 15.2; SAS Institute, Cary, NC, USA). The variables were expressed as mean ± standard deviation (SD), median (interquartile range [IQR], 25th–75th percentile), or the number of cases (%).

## 3. Results

In total, 169 patients with 172 UIAs who underwent 182 treatments were included in this study. The mean age of the patients was 61 ± 11 years, and 139 patients (83%) were female. The mean body mass index was 23 ± 3.9 kg/m^2^. Dual antiplatelet agents were prescribed in 153 patients (91%) (aspirin and clopidogrel: 125 (74%), aspirin and prasugrel: 28 (17%), clopidogrel and cilostazol: 1 (0.59%)), and 15 patients (8.9%) took triple antiplatelet agents (aspirin, clopidogrel, and cilostazol: 10 (5.9%), aspirin, prasugrel, and cilostazol: 5 (2.9%)). Forty-six patients (27%) received statins. Other patient characteristics are shown in Table 1.

The median maximum aneurysm size and neck length were 9.5 mm (IQR 6.1–14 mm) and 6.2 mm (IQR 4.4–8.0 mm), respectively. There were 94 aneurysms (52%) greater than or equal to 10 mm in diameter, and 88 (48%) were less than 10 mm. The aneurysms were located in the internal carotid artery (*n* = 144, 84%), vertebral artery (*n* = 14, 8.1%), basilar artery (*n* = 7, 4.1%), posterior communicating artery (*n* = 4, 2.3%), and middle cerebral artery (*n* = 3, 1.7%). Among the 145 saccular aneurysms, 97 were a side-wall type, and 48 were bifurcation type aneurysms. The median procedure time was 70 (IQR 60–84 min). Sixty-six aneurysms (38%) underwent balloon angioplasty in order to achieve better FD wall apposition. Other aneurysm characteristics are shown in Table 2.

### Outcome 

One patient died due to an intraoperative aneurysm rupture (0.59%). An mRS of 0–2 at 90 days was obtained in 164 of the 168 patients (98%). Steno-occlusive events of FD were observed in four patients (2.4%), and none of them had an increase in NIHSS score (Table 3). Among these four patients, two underwent medical treatment, one underwent mechanical thrombectomy, and on underwent stent placement. Other evaluated outcomes are shown in Table 3.

Conventional angiography, magnetic resonance angiography, and computed tomography angiography were used for 105 (61%), 64 (37%), and 3 (1.7%) aneurysms, respectively.

## 4. Discussion

Our study demonstrated that FD placement under LA is a safe and effective treatment for UIAs, with satisfactory occlusion and comparably low rates of permanent neurological morbidity and mortality. Furthermore, the procedure time was shorter than that in previous studies performed under GA, with a median procedure time of 70 (IQR 60-84) min (Table 4) [13,14,15,16,17,18,19,20,21,22,23,24,25]. Additionally, thromboembolic and hemorrhagic complications were comparable to the results of studies performed under GA (Table 4) [13,14,15,16,17,18,19,20,21,22,23,24,25].

FD has been shown to be effective in treating large, wide-necked, and anatomically challenging aneurysms [13,14,18]. Similar to other neuroendovascular procedures, FD placement is performed under GA for patient comfort, control of hemodynamics and respiration, and better image quality because of the decreased motion artifacts in immobile patients [4]. However, GA using an endotracheal tube is not risk-free and can cause a range of airway injuries, including hoarseness, vocal cord paralysis, tracheomalacia, and esophageal submucosal hematoma [26,27]. The incidence of airway complications caused by endotracheal intubation ranges from 0.5 to 7% [26]. Inhaled anesthetic agents have been shown to create significant vasodilation in the cerebral artery, which would give an inaccurate dimension of the vessel, causing the selection of inappropriately sized devices [28]. This can be avoided by performing the procedure under LA. Moreover, recent studies have shown technical success as well as a reduction in endovascular-related complications in patients undergoing carotid angioplasty and stenting and thrombectomy treated with LA [5,29,30].

Moreover, Rangel-Castilla et al. demonstrated the safety and feasibility of FD placement under conscious sedation (CS) for the treatment of intracranial aneurysms [25]. In addition, a retrospective study using a matched-cohort study design conducted by Griessenauer et al. found that CS was equivalent to GA in terms of occlusion rate, functional outcome, and complications [24].

The procedure under LA also allows for the serial monitoring of neurological function, especially at the time of microcatheterization or FD deployment, enabling the early detection of embolization or branch occlusion. Potential neurological changes precede any angiographic evidence of thromboembolism or hemorrhagic episodes [25]. These possible neurological complications would go undetected if the patient were treated with GA. Although intraprocedural neuromonitoring under GA is helpful, it is not as effective and accurate as direct neurological monitoring under LA. Therefore, the early detection of neurological deficits when the procedure is performed under LA could aid in rapid and accurate management and complete recovery, which would likely have been delayed under GA [25]. It is also important to carefully monitor the patient undergoing the procedure under LA who may require sedation as there might be difficulties in the neurological assessment of these patients.

A reduction in procedure time in FD is also an important advantage of LA over GA. Griessenauer et al. demonstrated a significantly shorter procedure time in patients treated under conscious sedation than in those treated under GA (CS: 64 min [16–353 min]; GA: 83 min [25–147 min]; *p* = 0.03) [24]. Our study also demonstrated a median procedure time of 70 (60–84) min. This is shorter than that of most procedures performed under GA (Table 4). In addition, FD deployment under LA not only shortens the procedure time but also minimizes fluoroscopy time, resulting in less radiation exposure [25]. Moreover, the experience of surgeons and improvement in the deliverability of FD would further reduce the procedure length, fluoroscopy time, and radiation exposure.

LA can be beneficial in patients with comorbidities in which GA is contraindicated or associated with a high risk of mortality and morbidity. Other benefits of FD deployment under LA include cost-effectiveness owing to shorter hospital stays and higher patient satisfaction. Zanaty et al. demonstrated the safety and feasibility of PED treatment under monitored anesthesia care, where patients were discharged home 6 h after the procedure [31]. Additionally, there is a debate that LA carries a greater risk of complications (such as wire perforation causing hemorrhage or dissection) due to patient agitation and movements that impede catheter navigation and degrade image quality. However, this has not been proven in previous studies [32]. Moreover, one study has shown that the use of a modified head holder could minimize excessive head movements during the procedure, producing an image quality comparable to those treated under GA [25].

### Limitations

This study had several limitations. First, it is a single-institutional retrospective experience, and as such, the results may not be generalizable. Second, a direct comparison between patients treated for similar lesions using GA and LA was not performed. Third, the study lacked data on patients who were converted to conscious sedation from LA. This is particularly important when patients treated with LA may experience fatigue and discomfort, as well as when there is an urgent need to induce CS because the patient cannot cooperate. Finally, data on fluoroscopy time and radiation exposure were unavailable. Nevertheless, our results support the treatment of intracranial aneurysms with FD deployment under LA.

## 5. Conclusions

Our study suggests that FD placement under LA is a safe and effective treatment for cerebral aneurysms with satisfactory occlusion and comparably low rates of permanent neurological morbidity and mortality. This is associated with a shorter procedure length. It is not associated with additional thromboembolic or hemorrhagic complications compared with FD under GA. Continuous neurological examination throughout the procedure may aid in the rapid recognition of neurological changes related to procedural complications and prompt appropriate management.

## Figures and Tables

**Figure 1 brainsci-12-01076-f001:**
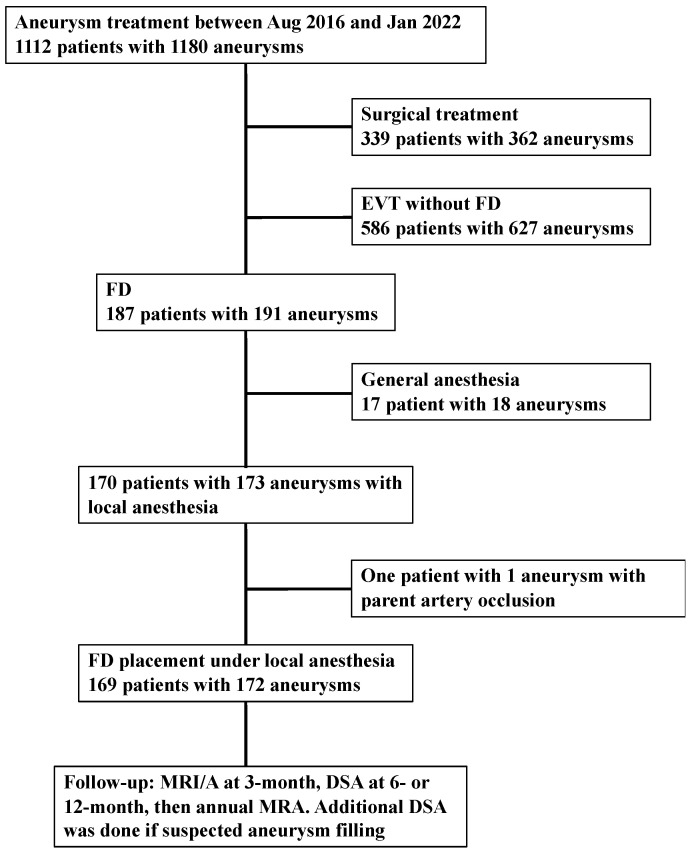
Flow chart of the selection process. Abbreviations: EVT = endovascular treatment; FD = flow diverter; MRI = magnetic resonance imaging; MRA = magnetic resonance angiography; DSA = digital subtraction angiography.

**Table 1 brainsci-12-01076-t001:** Clinical characteristics of 169 patients with cerebral aneurysms treated by FD under local anesthesia.

Variable	Value
Mean age, years (SD)	61 (11)
Female sex, *n* (%)	139 (82)
Smoking history, *n* (%)	64 (38)
Alcohol consumption, *n* (%)	32 (19)
Preoperative mRS 0–2, *n* (%)	165 (98)
Past medical history, *n* (%)	
Hypertension	68 (40)
Dyslipidemia	41 (24)
Diabetes mellitus	10 (5.9)
Verify Now, median (IQR)	
P2Y12 reaction units	184 (148–217)
Base	248 (219–286)
Inhibition	26 (8–41)
Aspirin reaction units	476 (410–523)
Platelet aggregability, median (IQR)	
Adenosine diphosphate	5 (4–5)
Collagen	4 (3–5)

Abbreviations: FD, flow diverter; IQR, interquartile range; mRS, modified Rankin scale; SD, standard deviation. Data expressed as number of cases (%), unless otherwise indicated.

**Table 2 brainsci-12-01076-t002:** Clinical and radiological characteristics of 172 aneurysms treated by FD under local anesthesia.

Variables	Value
Aneurysm shape, *n* (%)	
Saccular	145 (84)
Blister	3 (1.7)
Fusiform	24 (14)
Median maximum aneurysm size, mm (Q1–Q3)	9.5 (6.1–14)
Number of aneurysm measuring ≥10 mm (%)	94 (52%)
Median aneurysm neck, mm (Q1–Q3)	6.2 (4.4–8.0)
Growing aneurysm	47 (27)
Branch from aneurysm sac	7 (4.1)
Thrombosed aneurysm	15 (8.7)
Previous treatment of the same aneurysm, *n* (%)	26 (15%)
coiling	11 (6.4)
Stent assisted coiling	3 (1.7)
FD	7 (4.1)
Direct surgery	5 (2.9)
Number of FD treatment, *n* (%)	
1	162 (94)
2	8 (4.7)
3	1 (0.58)
Type of FD, *n* (%)	*n* = 181
FRED	103 (57)
PED	78 (43)
Balloon angioplasty, *n* (%)	66 (38%)

Abbreviations: FD = flow diverter; mRS = modified Rankin scale; FRED = Flow Re-Direction Endoluminal Device; PED = pipeline embolization device. Data expressed as number of cases (%), unless otherwise indicated.

**Table 3 brainsci-12-01076-t003:** Outcomes.

Variables	Value
Patient outcome	*n* = 169
mRS 0 to 2 at 90 days (*n* = 168)	164 (98)
Mortality at discharge	1 (0.59)
Major stroke	1 (0.59)
Steno-occlusive events	4 (2.4)
Aneurysm outcome	*n* = 172
Satisfactory aneurysm occlusion (*n* = 160)	126 (79)
Re-treatment	9 (5.2)
Post-treatment rupture	2 (1.2)

mRS = modified Rankin scale. Data expressed as number of cases (%), unless otherwise indicated.

**Table 4 brainsci-12-01076-t004:** Summary of results of the present study and previous studies.

Study	Year	Number of Aneurysm (Patient)	Anesthesia	Procedure Time (min)	Hemorrhagic Events	Ischemic Events	Aneurysm Occlusion	Neurologic Morbidity and Mortality
Current study	2022	172 (169)	LA without CS	70 (IQR 60–84)	0.59%	2.4%	79% (at median 345 days)	Mortality: 1.2%
PITA Nelson et al. [15]	2011	31 (31)	GA	NA	0%	6.7% (2/30)	93.3% (at 6 M)	Morbidity 6.7%
PUFS Becske et al. [13,16]	2013, 2017	109 (108)	GA	124	1.9% (2/107)	6.7% (2/30)	86.8 % (at 1Y) 95.2% (at 5 Y)	Morbidity 4.7% Mortality 2.8%
ASPIRe Kallmes et al. [17]	2014	207 (191)	NA	112.9 ± 54.9	3.7% (7/191)	1.6% (3/191)	74.8% (at 7.8 M)	Morbidity 7.4% Mortality 1.6%
IntePED Kallmes et al. [18]	2015	906 (793)	NA	101 ± 50	2.4% (19/793)	4.7% (37/793)	NA	Morbidity 7.4% Mortality 3.8%
Australian Registry Chiu et al. [19]	2015	119 (98)	NA	NA	2.4% (19/793)	4.7% (37/793)	93.2% (at 2 Y)	Mortality 0.8%
PREMIER Hanel et al. [20]	2019	141 (141)	GA or CS	78.4 ± 40.3	1.4% (2/141)	0.7% (1/141)	81.9% (at 1Y)	2.1%
Wakhloo et al. [21]	2015	190 (165)	GA (except in 4: CS)	79.8 (Average)	2.5%	3.7%	75% (at median, 6 M; range, 1–38)	Morbidity 6% Mortality 2.7%
Berge et al. [22]	2012	77 (65)	GA	NA	6.2%	4.7%	84.5% (at 12 M)	Morbidity 7.7% Mortality 3%
SAFE Pierot et al. [14]	2019	103 (103)	NA	NA	1%	6.8% (4/103)	73.3% (at 1 Y)	Morbidity 2.9% Mortality 1.9%
SCENT Myers et al. [23]	2019	180 (180)	NA	NA	2.8%	6.1% (11/180)	66.1% (at 1 Y)	Total 8.3% Mortality 2.8%
Griessenauer et al. [24]	2017	CS (199) GA (149)	CS (70 patients); GA (70 patients)	CS (64) GA (83)	5.7% (CS); 1.4% (GA)	5.7% (CS); 2.9% (GA)	75% (CS at 14 M); 82.4% (GA at 7 M)	Morbidity: 1.4% (CS); 3.2% (GA) Mortality: 1.4% (CS); 0% (GA)
Rangel-Castilla et al. [25].	2015	139 (130)	CS	85	NA	3%	NA	Morbidity 3%

Abbreviations: CS, conscious sedation; GA, general anesthesia; LA, local anesthesia; M, month; NA, not applicable; Y, year.

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
