# Peer review of "Clinical Results of Flow Diverter Treatments for Cerebral Aneurysms under Local Anesthesia"

_brainsci, 2022, doi:10.3390/brainsci12081076_

Round 1

Reviewer 1 Report

Thank you for this well written paper.

However, I have some major issues: 

-What was your follow-up regime? Please provide an additional figure with a time line (when DSA, when MRA?). You just state "satisfactory aneurysm occlusion" in table 3. What was the exact outcome data? How did you measure aneurysm occlusion on MRA? Modified Raymond-Roy occlusion classification? 

-Please also report outcome data on FDS wall adaption, configuration (e.g., fish-mouthing, foreshortening), endothelial thickening and in-stent thrombosis.

-Please state if and in how many cases there was a switch to general anesthesia and for what reasons.

-Line 103 ff. Who assessed neurological outcome? 

-Line 171: "In addition, FD deployment under LA not only shortens the procedure time but also 171 minimizes fluoroscopy time, resulting in less radiation exposure": this statement is not supported by your data as you do not provide fluoroscopy time/radiation dose.

-Line194: How is it possible that the data on radiation exposure is not available with regards to radioprotection requirements?

-There may be effects of general (inhalation) anesthetics on cerebral vessels as they act as vasodilators with regard the chosen device sizes. This may be mitigated/avoided if the procedure is performed in local anesthesia. Please discuss. DOI: 10.14744/tjtes.2021.00269

Reviewer 2 Report

In this article, the authors evaluated clinical results of flow diverter placement for unruptured cerebral aneurysms under local anesthesia, and reported that the procedure under local anesthesia was safe and effective. They provided awesome treatment results and well-written article; however, there are some concerns which should be resolved.

1.     According to the selection process (Table 1), 17 patients were treated their cerebral aneurysms under general anesthesia, and they were excluded from the analysis. I’m afraid that there was a selection bias because of the process. I think authors ought to indicate why some patients underwent treatments under general anesthesia and why they could be excluded. In addition, authors should show that 18 aneurysms had similar characteristics (aneurysm size, region of an aneurysm, patient factors), otherwise the conclusion will be applied only to the specific condition.

2.     Authors emphasized that treatments under local anesthesia would make it easy to detect neurological complications. However, neurological assessment sometimes becomes difficult when patients are injected some sedative drugs. In addition, patients occasionally come into restless states because of inappropriate use of drugs. Therefore, I think it is quite important to present the percentage of sedation usage. Although authors have already described it as a limitation, it will be worth to add the data if possible. If sedative drugs were rarely used, the present discussions are well-supported. If they were sometimes used, the unfavorable aspect of local anesthesia should be added in discussion. Furthermore, if you can, please provide the incidence of additional procedures due to the intraoperative neurological change, which will be also support your argument.

Reviewer 3 Report

Thank you for the article.

I think that for the exact conclusions, as a single centre retrospective study, there must be done a direct comparison between GA and LA, data about conversion and complications.

Round 2

Reviewer 1 Report

I believe the manuscript has been sufficiently improved

Reviewer 3 Report

Dear authors. 

It is true that you mentioned the issue in the limitations paragraph but without adequate comparisons it is not clear why you should do the procedure in LA and not GA. 

The selection of anesthesia is not a big issue. 
